# SmartRAG: Jointly Learn RAG-Related Tasks From the Environment Feedback

**Jingsheng Gao**$^{\diamond\S*}$**, Linxu Li**$^{\S}$**, Ke Ji**$^{\ddagger\S}$**, Weiyuan Li**$^{\S}$**, Yixin Lian**$^{\S}$**, Yuzhuo Fu**$^{\diamond\dagger}$**, Bin Dai**$^{\S\dagger}$
$^{\diamond}$ Shanghai Jiao Tong University    $^{\S}$Xiaobing.AI
$^{\ddagger}$ The Chinese University of Hong Kong, Shenzhen
`{gaojingsheng, yzfu}@sjtu.edu.cn  {keji}@link.cuhk.edu.cn`
`{lilinxu, liweiyuan, lianyixin, daibin}@xiaobing.ai`

## Abstract

RAG systems consist of multiple modules to work together. However, these modules are usually separately trained. We argue that a system like RAG that incorporates multiple modules should be jointly optimized to achieve optimal performance. To demonstrate this, we design a specific pipeline called **SmartRAG** that includes a policy network and a retriever. The policy network can serve as 1) a decision maker that decides when to retrieve, 2) a query rewriter to generate a query most suited to the retriever, and 3) an answer generator that produces the final response with/without the observations. We then propose to jointly optimize the whole system using a reinforcement learning algorithm, with the reward designed to encourage the system to achieve the best performance with minimal retrieval cost. When jointly optimized, all the modules can be aware of how other modules are working and thus find the best way to work together as a complete system. Empirical results demonstrate that the jointly optimized SmartRAG can achieve better performance than separately optimized counterparts.

## 1 Introduction

Although large language models(LLMs) (Chowdhery et al., 2023; Touvron et al., 2023; Chung et al., 2024) have demonstrated exceptional capabilities across various domains, addressing knowledge-related issues beyond model parameters remains a challenging task (Mallen et al., 2023b; Min et al., 2023). Retrieval-augmentation generation(RAG) effectively enhances model performance in these scenarios by retrieving additional information from external tools (Ram et al., 2023).

RAG systems usually consist of multiple modules including at least a retriever and a generator. Some systems may have other modules like a reranker (Glass et al., 2022), a decision maker deciding when to retrieve (Jeong et al., 2024; Wang et al., 2023), a query rewriter (Ma et al., 2023; Tan et al., 2024) or a verifier (Lewis et al., 2020; Izacard et al., 2023). These modules are often hand-designed and separately optimized. One of the issues is that the *golden answer* of the intermediate modules are usually not accessible. What is worse, sometimes the *golden answer* is model-dependent or retriever-dependent. For example, Asai et al. (2024) uses the result of GPT4 (Achiam et al., 2023) as the ground truth for the decision maker, which can be suboptimal. A question that GPT4 can answer without retrieval may need retrieval for other base models, meaning that the *golden answer* for the decision maker is model-dependent. In this sense, optimizing each module separately will obviously be suboptimal.

We argue that a system like RAG that incorporates multiple modules should be jointly optimized in an end-to-end manner. For this purpose, we design a RAG system, namely SmartRAG, and jointly optimize it using reinforcement learning, with environment feedback as supervision.

Specifically, SmartRAG includes two core building blocks, a policy network, and a retriever, as shown in Figure 1. The policy network can serve as three different roles. Firstly, it should decide

---

$^{*}$ Work done during an internship at Xiaobing.AI
$^{\dagger}$ Corresponding Author

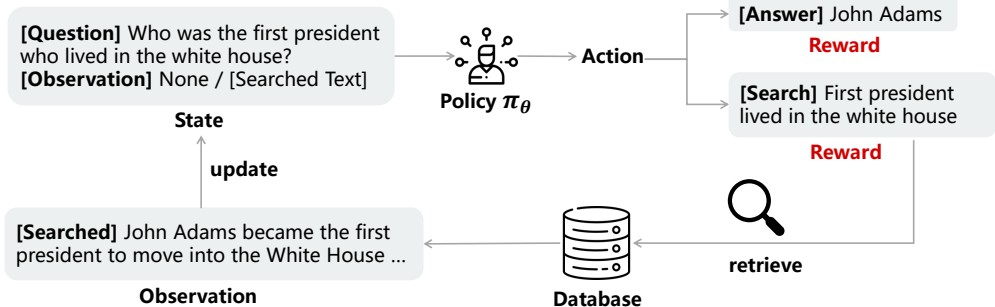

Figure 1: Overview of SmartRAG. The policy network takes the original question and optional observations as input and generates an action that can be interpreted as a decision-maker and the corresponding parameters. If the policy decides to retrieve, the retriever will be called to obtain the observation that updates the state. Otherwise, the policy network will output the final answer. SmartRAG is optimized using a reinforcement learning algorithm with a properly designed reward.

whether to retrieve or not based on the input state, which includes the input question and optional observations. The second role is query rewriter. If the policy network decides to retrieve the database, it will also output the search query that better suits the retriever than the original question. The last role is answer generator. If the policy network thinks it is already able to answer the question based on the current information, it should generate an appropriate final response.

To optimize such a system, the only supervision should come from the environment, rather than a proxy "golden" answer generated by another model. For SmartRAG, the primary goal is to correctly answer the question while the secondary goal is to minimize the retrieval cost. With this guideline, we can design a reward function that encourages the whole system to correctly answer the question with minimal retrieval trials.

We validate the effectiveness of SmartRAG on various datasets, demonstrating that our SmartRAG outperforms its counterparts with separately optimized modules. We also systematically analyze the properties of SmartRAG, showcasing that it learns *when to retrieve, what to retrieve* and *how to answer based on the observations*. The integration of these competencies necessitates a high degree of inter-module awareness, which in turn enhances the system's overall performance.

Our contribution lies in three aspects:

1. We design SmartRAG, a system that includes a policy network and a retriever. The policy network can serve as three different roles related to RAG. With this design, our SmartRAG system can be optimized using an RL framework.

2. We propose using PPO to jointly optimize our SmartRAG to achieve better performance than training each related module separately.

3. Empirical results demonstrate the effectiveness of our method on various tasks. Systematic analysis are also applied to help us better understand how SmartRAG works and why it outperforms previous algorithms.

## 2 RELATED WORK

### 2.1 RETRIEVAL-AUGMENTED GENERATION

Retrieval augmentation techniques have been utilized to acquire external knowledge, which aids language models in achieving superior performance across a broad spectrum of tasks (Guu et al., 2020; Gao et al., 2023; Hu & Lu, 2024). Previous research has primarily concentrated on "what to retrieve"(Khandelwal et al., 2019; Ram et al., 2023) and "how to utilize the retrieved information"(Khattab et al., 2022).

Recently, some studies have begun to explore when to use retrieval to meet the varying requirements of diverse tasks (Jiang et al., 2023). For instance, Mallen et al. (2023b) assesses the popularity

of a query based on the frequency of its entities and recommends activating retrieval modules only when the entity frequency drops below a predetermined threshold. Wang et al. (2023) enhances the model's retrieval performance by using self-knowledge to decide what they know and do not know. Self-Rag (Asai et al., 2024) leverages special tokens to adaptively retrieve external knowledge and confirm the output's relevance or efficacy. Additionally, some researchers incorporated supplementary modules or classifiers to determine whether a query necessitates additional knowledge for resolution (Liu et al., 2024; Jeong et al., 2024). Wang et al. (2024) create a compositional unknown dataset (CuQA), and utilize the confidence words or scores to decide whether to retrieve.

Unlike previous methods, we have implemented an end-to-end approach to jointly optimize all the related tasks such that the whole system can achieve better performance.

## 2.2 LEARNING FROM FEEDBACK

Training LLMs from feedback (Ouyang et al., 2022; Ji et al., 2023) has proven effective in enabling LLMs to understand the impact of their actions and adapt their behavior accordingly, which effectively aligns the model's behavior with human intentions. Based on the different forms of explicit guidance, it can be categorized into four types: label-based (Guerdan et al., 2023), reward-based (Wu et al., 2024), demonstration-based (Dasari et al., 2023), and comparison-based approaches (Ouyang et al., 2022). One of the most effective training paradigms among them is reinforcement learning (Ge et al., 2024). The significant reason is that it is relatively easier to collect and evaluate the quality of responses compared to expert answers (Rafailov et al., 2024), improving proficiency in translation (Kreutzer et al., 2018), summarization (Liu et al., 2020), and instruction-following Ramamurthy et al. (2023). In retrieval augmentation, Ma et al. 2023 employs Proximal Policy Optimization (PPO) (Schulman et al., 2017b) to refine the search query, using the match score between the predicted answer and the gold standard answer as the reward. Furthermore, this use of gold answers as feedback enables the model to learn when to utilize tools to expand its capabilities and interact with the real world (Qiao et al., 2024).

Different from most of the previous methods, which are solving a context bandit problem, our proposed SmartRAG contains multiple steps of interactions between the policy and the environment and learns from the environment feedback rather than human feedback.

## 3 METHODOLOGY

Let $\mathbf{x}$ be the input question and $\mathbf{y}$ be the golden answer. For RAG systems, there is usually a companion retriever $\mathcal{R}$ that takes a query $\mathbf{q}$ as input and returns an observation $\mathbf{o}$, *i.e.* $\mathbf{o} = \mathcal{R}(\mathbf{q})$.

The core part of SmartRAG is a policy network $\pi_\theta$, which is an LLM with the parameters being $\theta$. The policy network takes the state $\mathbf{s}$ as input, which is a combination of the original question and the optional observations, *i.e.*

$$\mathbf{a} \sim \pi_\theta(\mathbf{s} = [\mathbf{x}, \mathbf{os}]), \tag{1}$$

where $\mathbf{os}$ is the concatenation of all the historical observations returned from the retriever while $\mathbf{a}$ is the action sampled from the output of the policy network. In SmartRAG, there are two different types of actions: *answer* and *retrieve*. Let $\mathbf{a}_0$ be the first token of $\mathbf{a}$. Then $\mathbf{a}_0$ is a special token that can be either [ANSWER] or [RETRIEVE] . If $\mathbf{a}_0 = $ [RETRIEVE], it means that the policy thinks it necessary to retrieve with query $\mathbf{a}_{1:}$, which is the rest part of $\mathbf{a}$. In this case, the retriever will be called to obtain the new observation $\mathcal{R}(\mathbf{a}_{1:})$ that will be appended to $\mathbf{os}$. Then the policy network will be called again to produce the next action by (1). On the other hand, if $\mathbf{a}_0 = $ [ANSWER], it indicates that the policy network thinks there is no need for an extra retrieve and it is able to produce the right answer with $\mathbf{a}_{1:}$. Thus we obtain the final answer $\hat{\mathbf{y}} = \mathbf{a}_{1:}$.

The SmartRAG pipeline is shown in Algorithm 1. We can set the quota for retrieval as $N$. When the number of retrieves has already met the quota, we can force the policy network to generate the answer by forcing $\mathbf{a}_0$, which is the first output token, to be [ANSWER] . Thus we can avoid the policy network to keep retrieving if no satisfying observation is obtained from the retriever. Of course the retrieval quota $N$ is not a mandatory setting. We can remove it if unnecessary in certain applications.

---

**Algorithm 1** SmartRAG Pipeline

---

**Require:** Policy Network $\pi_\theta$, Retriever $\mathcal{R}$
1: **Input:** input question $\mathbf{x}$, retrieve quota $N$, observation $\mathbf{os} \leftarrow [\,]$, retrieve count $n \leftarrow 0$
2: **while** $n \leq N$ **do**
3:     **if** $n = N$ **then**,
4:        $\mathbf{a} \sim \pi_\theta\left([\mathbf{x}, \mathbf{os}]\right)$    s.t.    $\mathbf{a}_0 = $ **[ANSWER]**
5:     **else**
6:        $\mathbf{a} \sim \pi_\theta\left([\mathbf{x}, \mathbf{os}]\right)$
7:     $n \leftarrow n + 1$
8:     **if** $\mathbf{a}_0 = $ **[RETRIEVE]** **then**,
9:        $\mathbf{o} \leftarrow \mathcal{R}(\mathbf{a}_{1:})$,    $\mathbf{os} \leftarrow [\mathbf{os}, \mathbf{o}]$
10:    **else**
11:       return $\mathbf{a}_{1:}$

---

## 3.1 WARM-UP USING SUPERVISED FINETUNING

Since our policy network is initialized using a base LLM, the output of which does not necessarily follow our design that the first output token is either **[RETRIEVE]** or **[ANSWER]** . To achieve the output format requirements, we first warm-up the base LLM with our constructed dataset.

For each question $\mathbf{x}$, we construct three different types of input-output pairs for the warm-up step. The first pair is a direct answer. In this case, the input is the original question $\mathbf{x}$ while the output is the golden answer prefixed with the special token **[ANSWER]** . Thus we have the first data point, *i.e.*

$$\mathbf{s} = [\mathbf{x}, \mathbf{os} = [\,]], \quad \mathbf{a} = [\textbf{[ANSWER]}, \mathbf{y}]. \tag{2}$$

The second data point encourages the policy network to apply retrieval action. To achieve a diverse initial state for the future RL algorithm, we ask GPT-3.5 [1] to rewrite the original question $\mathbf{x}$ to produce a diverse query for the retriever. The second data point then becomes

$$\mathbf{s} = [\mathbf{x}, \mathbf{os} = [\,]], \quad \mathbf{a} = [\textbf{[RETRIEVE]}, \text{GPT-Rewriter}(\mathbf{x})] \tag{3}$$

The third data point enables the initial policy network to know how to answer with the observations, it is constructed as

$$\mathbf{s} = [\mathbf{x}, \mathbf{os} = \mathcal{R}(\text{GPT-Rewriter}(\mathbf{x}))], \quad \mathbf{a} = [\textbf{[ANSWER]}, \mathbf{y}]. \tag{4}$$

After finetuning the base LLM with the above-constructed dataset, we can obtain an initial policy network $\pi_{\theta_0}$ that can output with our desired format and generate reasonable results. It should be noticed that the warm-up step only provides a reasonable initial policy. It does not necessarily mean that the constructed dataset from (2) to (4) is the "*golden*" answer since the behavior of the policy network will be further optimized in the following reinforcement learning step.

We can further enhance the initial policy by slightly modifying the warm-up dataset. For the data points that the base LLM can already correctly answer, we remove the constructed SFT data points defined in (3) and (4). Otherwise, we remove the constructed SFT data point defined in (2). By doing so, the warm-up dataset already contains the information about whether the base LLM has the corresponding knowledge for the question, leading to a better initial policy, denoted as $\pi_{\theta_0^*}$.

## 3.2 JOINTLY OPTIMIZATION USING REINFORCEMENT LEARNING

After obtaining the initial policy $\pi_{\theta_0^*}$ in the SFT warm-up stage, we then start jointly optimizing the whole framework using reinforcement learning. Following Algorithm 1, we can generate a trajectory $(\mathbf{s}^{(0)}, \mathbf{a}^{(0)}, \mathbf{r}^{(0)}, ..., \mathbf{s}^{(T)}, \mathbf{a}^{(T)}, \mathbf{r}^{(T)})$, where $\mathbf{s}^{(t)}$, $\mathbf{a}^{(t)}$ and $\mathbf{r}^{(t)}$ represents the state, action and reward at the $t$-th step respectively. $T$ stands for the trajectory length and we then define the details of the reward $\mathbf{r}^{(t)}$.

---

[1] gpt-3.5-turbo-1106

Our reward function is only related to the action $\mathbf{a}^{(t)}$. If the policy produces a related answer, the system will get a positive reward. If the policy uses a retrieval opportunity, it should be penalized for the retrieval cost. Thus the reward function can be written as

$$
\mathbf{r}^{(t)} = \begin{cases} \mathbb{I}\left(\mathbf{a}_{1:}^{(t)}, \ \mathbf{y}\right) + F_1\left(\mathbf{a}_{1:}^{(t)}, \ \mathbf{y}\right) & \text{if } \ \mathbf{a}_0^{(t)} = [\text{ANSWER}], \\ -\alpha & \text{if } \ \mathbf{a}_0^{(t)} = [\text{RETRIEVE}]. \end{cases} \tag{5}
$$

Here $\mathbb{I}$ is the identity function. It equals to $1$ if the two arguments are identical and equals to $0$ otherwise. Besides the exact match reward term, we also add a soft match term in $\mathbf{a}_0^{(t)} = [\text{ANSWER}]$ case, which is the $F_1$ score between $\mathbf{a}_{1:}^{(t)}$ and $\mathbf{y}$. In addition, $\alpha$ is a positive constant penalizing the retrieval cost. In some applications, where retrieval cost is not an issue, we can simply set $\alpha$ to be $0$ to remove the penalty. The overall objective then becomes

$$
\mathbb{E}_{\pi_\theta}\left[\sum_{t=0}^{T} \gamma^t \cdot \mathbf{r}^{(t)}\right], \tag{6}
$$

where $\gamma$ is the discount factor.

By such a design, we encourage the whole SmartRAG system to produce the correct answer with a minimal number of retrieval trials. As a result, the system will only retrieve when the model does not inherently know the answer to the question while the retriever happens to have the related knowledge. The system will not call the retriever when the base LLM already knows the answer or neither the base LLM nor the retriever does not know the correct answer. All the modules in the SmartRAG pipeline work as a whole system rather than as separate independent parts.

We optimize the objective (6) using proximal policy optimization (PPO) (Schulman et al., 2017a).

## 4 EXPERIMENTS

**Dataset.** Following Ma et al. (2023), we use three open-domain QA datasets for evaluation, namely PopQA (Mallen et al., 2023a), AmbigNQ (Min et al., 2020) and HotpotQA (Yang et al., 2018). Besides, we also use other datasets like TrivialQA (Joshi et al., 2017), OpenBookQA (Mihaylov et al., 2018), MedQA-cn (Jin et al., 2021) and ARC-c (Clark et al., 2018) for more analysis and comparisons. All the details of the related datasets are shown in Appendix A.1.

**Retriever.** We utilize Bing search engine as our retriever $\mathcal{R}$, which offers a broad knowledge scope and ensures up-to-date factual information. The search engine will return multiple results given a query. We select the top $K$ results and concatenate the snippets from these results as the observation. We leave how to utilize these snippets to the answer generator. In our experiments, we set $K = 4$ without other clarifications.

**Training Details.** In the SFT warmup stage, we finetune the network for $1$ epoch with the learning rate being $3e - 4$. The batch size is set to be $8$. In the reinforcement learning stage, we use an on-policy sampling strategy. For each iteration, we sample many trajectories such that the total length of all these trajectories is $5120$. The policy network is then trained for $1$ epoch on these samples with the learning rate being $2e - 6$ and batch size being $32$. The retrieval penalty $\alpha$ in (5) is set to $0.2$ and the retrieval quota $N$ is specified to be $1$ to fairly compare with other baselines.

More implementation details are introduced in Appendix A. We compare our method with the following baselines.

**Vanilla RAG** retrieves the search engine using the input question. Then it asks the LLM to generate the final answer with the observations returned from the search engine in a few-shot manner.

**SFT RAG** uses the same pipeline as our SmartRAG. Similar to Vanilla RAG, it uses the input question to retrieve the search engine. We then finetune the answer generator given the input question along with the observations. SFT RAG can be regarded as only training the answer generator and leave the other two modules as they are.

**GPT4 Rewriter + SFT RAG** adds a query rewrite module than **SFT RAG**. We ask GPT4, which is one of the most powerful LLMs, to rewrite the search query before feeding it to the search engine. In this case, the answer generator and the query rewriter are separately optimized.

| Model | PopQA | | AmbigNQ | | HotpotQA | | Average | |
|---|---|---|---|---|---|---|---|---|
| | EM | F1 | EM | F1 | EM | F1 | EM | F1 |
| *Flan-T5 large* | | | | | | | | |
| ICL | 7.03 | 10.16 | 2.47 | 7.30 | 12.46 | 20.42 | 7.32 | 12.63 |
| SFT | 12.76 | 17.70 | 4.56 | 10.92 | 12.72 | 19.72 | 10.01 | 16.11 |
| Vanilla RAG | 34.36 | 38.80 | 30.72 | 40.16 | 22.06 | 32.57 | 29.05 | 37.18 |
| SFT RAG | 38.15 | 42.05 | 31.63 | 41.05 | 24.01 | 34.03 | 30.93 | 38.68 |
| GPT4 Rewriter + SFT RAG | 39.27 | 42.68 | 33.43 | 42.40 | 24.12 | 34.33 | 32.27 | 39.80 |
| **SmartRAG(ours)** | **42.50** | **45.95** | **33.71** | **42.75** | **25.54** | **35.23** | **33.29** | **41.25** |
| *LlaMa-2 7B* | | | | | | | | |
| ICL | 21.79 | 24.61 | 18.63 | 27.12 | 17.62 | 25.60 | 19.35 | 29.11 |
| SFT | 27.91 | 31.54 | 22.47 | 31.28 | 20.74 | 29.52 | 23.71 | 30.78 |
| ReAct (Yao et al., 2023) | 27.21 | 31.49 | 22.45 | 31.24 | 21.91 | 29.83 | 23.86 | 30.85 |
| IRCoT (Trivedi et al., 2023) | 9.22 | 17.93 | 13.52 | 25.94 | 17.42 | 27.13 | 13.39 | 23.67 |
| Self-RAG (Asai et al., 2024) | 1.07 | 23.49 | 6.12 | 26.48 | 6.80 | 17.53 | 4.66 | 22.50 |
| SKR (Wang et al., 2023) | 36.04 | 39.59 | 35.41 | 44.75 | 23.63 | 33.44 | 31.69 | 39.26 |
| GPT4 Rewriter + SKR | 39.25 | 42.94 | 35.03 | 44.68 | 23.39 | 33.61 | 32.73 | 40.41 |
| Vanilla RAG | 32.26 | 40.06 | 33.12 | 46.20 | 24.18 | 34.94 | 29.82 | 40.40 |
| SFT RAG | 38.71 | 42.33 | 36.58 | 46.09 | 25.59 | 35.15 | 33.64 | 41.19 |
| GPT4 Rewriter + SFT RAG | 42.22 | 46.04 | 36.13 | 46.02 | 25.29 | 35.38 | 34.54 | 42.49 |
| **SmartRAG(ours)** | **44.32** | **47.60** | **37.89** | **47.76** | **26.60** | **36.42** | **36.27** | **43.93** |

Table 1: Performance on open-domain QA datasets. Bold numbers indicate the best result while underlined number indicates the second-best result.

**Self-RAG** (Asai et al., 2024) leverages GPT-4 to construct a large-scale dataset encompassing tasks such as determining whether retrieval is necessary, assessing the relevance of retrieved content, and formulating appropriate responses. Subsequently, SFT is employed to enable the LLM to adaptively perform retrieval and generation. Retrieval is triggered when the model predicts a special retrieval token exceeding a predefined threshold, followed by answer generation.

**SKR** (Wang et al., 2023) is also an adaptive retrieval algorithm that utilizes the model's correctness in responding to the training set as a supervisory signal: correct responses indicate that retrieval is unnecessary, while incorrect responses suggest the need for retrieval. Based on this, a classifier is then trained to predict whether a sample requires retrieval.

**ReAct** (Yao et al., 2023) uses the inherent ability of the base network to decide how to retrieve and answer using chain-of-thought (CoT) (Wei et al., 2022). It can also self-reflect the previous queries and answers to arrive at the final answer.

**IRCoT** (Trivedi et al., 2023) enhances each step of the CoT generation process by incorporating knowledge retrieval steps during the generation process. Note that both ReAct and IRCoT use multi-step retrieval such that the retrieval cost is higher than other baselines and our SmartRAG.

All the above baselines use at least one time of retrieve. To further show the baseline without any retrieval, we also compare with two extra baselines, namely **ICL** (In-Context Learning) which is simply a few-shot inference, and **SFT** which is directly finetuned on the golden answer.

Following Jeong et al. (2024); Asai et al. (2024), we use two different types of base LLMs, *i.e.* Flan-T5 large (Chung et al., 2024) and LlaMa2-7B (Touvron et al., 2023). Two different metrics are adopted in the comparison: the exact match (EM) and the F1 score.

## 4.1 ADVANTAGE OF JOINT OPTIMIZATION

We compare our proposed SmartRAG with all the baselines in Table 1. In this table, **ICL** and **SFT** show the results without retrieval. The performance is very poor, indicating that retrieving

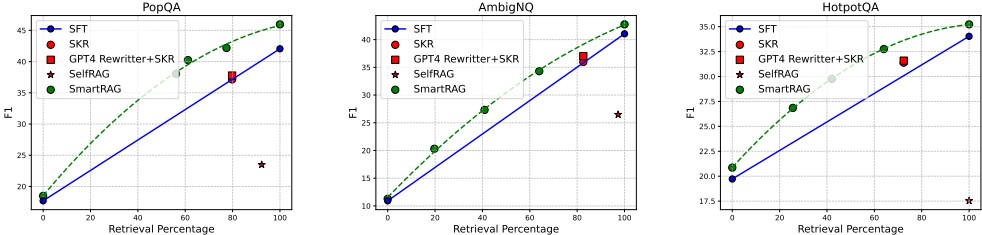

Figure 2: F1 Score of different retrieval percentage across three datasets on Flan-T5-large.

is necessary for these datasets. All the algorithms that have retrieve-related modules significantly increase the performance.

By jointly optimizing all the related modules, our SmartRAG is able to outperform all the counterparts that use separately optimized modules. This validates our argument that joint optimization is necessary to make all the modules properly work together to achieve optimal performance.

Specifically, **SFT RAG** only trains the answer generator while **GPT4 Rewriter + SFT RAG** separately learns both the answer generator and the query rewriter. Both methods performs worse than **SmartRAG** on all the datasets. In addition, we find that **GPT4 Rewriter + SFT RAG** sometimes performs worse than **SFT RAG**, meaning that the rewritten query provided by GPT4 may be worse than the original question. In this sense, the query rewritten by GPT4 is just *reasonable* but not *golden*.

Compared to **SFT RAG**, **SKR** further learns the decision maker to decide whether to retrieve, thereby reducing retrieval costs while incurring a degradation in performance. **GPT4 Rewriter + SKR** separately learns all three modules and thus performs better than vanilla **SKR**. Yet both EM and F1 score are worse than **SmartRAG**.

We only provide the result of **ReAct** and **IRCoT** for the LlaMa2 base model because these two methods produce extremely poor results on Flan-T5. Without any adaptation to the real application, which includes tasks that are either model-dependent or retriever-dependent, simply using the inherent reasoning ability of the base models has no means of achieving optimal performance.

## 4.2 SMARTRAG LEARNS WHEN TO RETRIEVE

The first token of the action **a** indicates whether to retrieve or not. In practice, the policy network outputs a Bernoulli distribution over $\{[\text{ANSWER}], [\text{RETRIEVE}]\}$, we can manually set up a threshold on the logit of $[\text{ANSWER}]$ to control the retrieval ratio. Figure 2 shows the F1 score of SmartRAG with different thresholds. We compare SmartRAG with four baselines. The first one is SFT. Since it does not have any decision maker for whether to retrieve, it only has two states: full retrieve and no retrieve. The other three baselines are **SKR** (Wang et al., 2023), **GPT4 Rewriter + SKR** and **Self-RAG** (Asai et al., 2024). We can see that the results of SmartRAG is to the upper left side of the other methods, meaning that SmartRAG can produce a similar F1 score with less retrieval cost. On both PopQA and HotpotQA datasets, SmartRAG can even produce a higher F1 score with less retrieval percentage than SKR. To achieve this, SmartRAG should put more retrieval efforts on the scenarios when retrieval does bring benefit. These are instances where the base LLMs lack inherent knowledge of the correct answer, yet the necessary information is encompassed within the database.

A deduction about the above arguments is that SmartRAG should not retrieve anything if the database hardly contains any useful knowledge. This is indeed the case for the Bing search engine on OpenBookQA, MedQA-cn and ARC-c datasets. We list the retrieval percentage and accuracy on these datasets in Table 2. SFT and SFT RAG produce extremely close accuracy since the database contains very little knowledge of these datasets. The SKR method still produces a high retrieval percentage, which wastes the retrieval resources. Our SmartRAG is aware of the property of the database and learns not to retrieve it in these cases.

| Method | Retrieval Percentage ↓ | | | Accuracy ↑ | | |
|---|---|---|---|---|---|---|
| | OpenBookQA | MedQA-cn | ARC-c | OpenBookQA | MedQA-cn | ARC-c |
| SFT | 0.00 | 0.00 | 0.00 | 67.20 | 31.66 | 54.95 |
| SFT RAG | 100 | 100 | 100 | 68.20 | 31.11 | 56.99 |
| SKR (Wang et al., 2023) | 5.60 | 84.29 | 9.98 | 67.20 | 31.89 | 54.86 |
| SmartRAG | 0.00 | 0.00 | 0.00 | 66.60 | 33.31 | 55.72 |

Table 2: Results of different one-choice questions on Flan-T5-large. All the methods produce similar accuracy since the database does not contain the useful answer. Our SmartRAG learns not to retrieve for it does not bring any benefit in these cases.

---

**Question 1:** Who was the producer of 9?
**Golden Answer:** Shane Acker
**Observation with original question:** 5 Things to Know About THE9, China 2019s New Pop Idol Group: Predicting the shelf-life of a pop act is never straightforward, but it will be interesting to see just how long THE9 stick around given the rapid demise of boyband Nine Percent, who were spawned by 2018-2019s Idol Producer.
**Output:** [ANSWER] Tim Burton

**Rewritten Query:** Who produced the movie 9?
**Observation with rewritten query:** 9 (2009) - IMDb: Shane Acker first made 9 (2009) as a ten minute short film while he was still at UCLA. It was nominated for Best Animated Short at the Oscars, and although it didn't win, Acker was offered the chance to expand it into a feature film.
**Output:** [ANSWER] Shane Acker

**Question 2:** When was the contruction of the makkah royal clock tower hotel started?
**Golden Answer:** 2004
**Observation with original question:** Ryugyong Hotel: Ryugyong Hotel was designed by North Korean firm Baikdoosan Architects and Engineers, which is also the contractor of the project before 1992, the firm had built the building to its full height before the construction suspended in 1992. ... However, considering Ryugyong Hotel is 112 meters shorter than Willis Tower, 360,000 square meters is still ...
**Output:** [ANSWER] 1992

**Rewritten Query:** Start date of construction for the Makkah Royal Clock Tower Hotel
**Observation with rewritten query:** Makkah Royal Clock Tower, Saudi Arabia - World Construction Network: The Makkah Royal Clock Tower complex, also known as the Abraj Al-Bait Towers, located near Masjid al Haram in Mecca, Saudi Arabia, is a mixed residential and hotel complex. Construction on the complex was started in 2004 and finished in 2012.
**Output:** [ANSWER] 2004

Table 3: Comparison between observations with orignal questions and rewritten queries.

## 4.3 SMARTRAG LEARNS WHAT TO RETRIEVE

Once SmartRAG decides to retrieve the database, the next task is *what to retrieve*. Directly searching the original question produces suboptimal performance, as Table 1 already demonstrates. In this section, we showcase how SmartRAG rewrites the retrieving query in Table 3 with two examples. In the first question *Who was the producer of 9*, directly searching the question cannot provide any useful information because the term *9* is quite confusing. As a result, the retriever does not return any useful information and thus the final answer is wrong. The rewritten query *Who produced the movie 9* is aware that *9* is a movie name and explicitly uses the term *movie 9* to retrieve. With this modification, the returned observation is more related and contains the correct answer, which is then output by the answer generator in the next step. In the second example, the subject of the original question is *makkah royal clock tower hotel*. Directly searching the question returns information about the hotel with no messages related to the start date of construction for the hotel. The subject is modified to *start date of the construction* by the policy network, making the search more accurate.

To further validate that the query rewrite module does play its role in enhancing the overall performance, we replace all the retrieving queries in SmartRAG with the original question. The evaluation result is shown in Table 4 (comparing lines *SmartRAG* and *Replace Query*). Replacing the rewritten query in SmartRAG with the original question does degrade the performance on all the datasets over both exact match and F1 score metrics. We also introduce a new metric *hit*. It tracks the ratio of observations that contain the golden answer.

| Method | PopQA | | | AmbigNQ | | | HotpotQA | | |
|---|---|---|---|---|---|---|---|---|---|
| | EM | F1 | hit | EM | F1 | hit | EM | F1 | hit |
| Replace Query | 39.13 | 42.98 | 50.49 | 33.41 | 42.19 | 49.31 | 24.31 | 34.04 | 33.83 |
| Replace Generator | 40.95 | 43.87 | **52.59** | 33.23 | 42.02 | **49.78** | **25.63** | 35.04 | **35.89** |
| **SmartRAG** | **42.50** | **45.95** | **52.59** | **33.71** | **42.75** | **49.78** | 25.54 | **35.23** | **35.89** |

Table 4: Comparison to replace the query with the original questions and replace the generator with the pre-RL warm-up model.

---

**Question 1:** Who was the screenwriter for On the Beach?
**Golden Answer:** Nevil Shute
**Observation:** On the Beach, Summary and Facts, Britannica - Encyclopedia Britannica: On the Beach, American dramatic film, released in 1959, that was set in the aftermath of an imagined World War III. It was based on the apocalyptic novel of the same name by Nevil Shute. The fatal fallout of nuclear war in the year 1964 serves as the fictional backdrop for romance between a navy On the Beach (1959) - FilmAffinity: On the Beach is a film directed by Stanley Kramer with Gregory Peck, Ava Gardner, Fred Astaire, Anthony Perkins ....

**Before PPO Output:** [ANSWER] Stanley Kramer

**After PPO Output:** [ANSWER] Nevil Shute

---

**Question 2:** What's the most points scored in an nba game by a single team?
**Golden Answer:** 186
**Observation:** NBA Single Game Leaders and Records for Points: Checkout the complete list of NBASingle Game Leaders and Records for and more on Basketball-Reference.com. Year-by-Year Top 10: Checkout the complete list of NBA and ABAYear-by-Year Top 10 Leaders and Records for and more on Basketball-Reference.com ... shows the teams that scored the most points in one game in NBA history, including regular season and Playoffs. Most points scored by a team in an NBA game: On Dec. 13, 1983, the Detroit Pistons set the single-game scoring record for an NBA team with 186 points in an all-time shootout against the Nuggets. Detroit defeated Denver 186-184 in... Who has scored the most points in a single NBA game? - ESPN: On March 2, 1962, Wilt Chamberlain set the NBA's single-game scoring record with 100 points in the Philadelphia Warriors' 169-147 win over the New York Knicks. The NBA didn't add the...

**Before PPO Output:** [ANSWER] 370

**After PPO Output:** [ANSWER] 186

Table 5: Improvement of answer from observation between before and after PPO.

## 4.4 SMARTRAG LEARNS HOW TO ANSWER

Given the correct time to retrieve and the proper query for retrieval, the system should also learn how to well utilize the observations from the retriever to produce the final answer. To validate that SmartRAG learns how to answer with the observations, we replace the final answer generation step with the version before PPO. Note that this generator before PPO is already finetuned with data points that contain observations from the retriever. So it also learns how to answer based on the observations. The only difference is that it is not jointly optimized with other modules.

Table 4 (comparing lines *SmartRAG* and *Replace Generator*) shows the replacement also degrades the performance on most of the metrics. The *hit* metric remains the same because they use the same queries for retrieving and thus have exactly the same observations. Two examples are presented in Table 5. In both examples, the correct answer is included in the observations (highlighted in green). SmartRAG can correctly find the answer from the lengthy information. However, the answer generator before PPO either finds the wrong information (the first example) or hallucinates the wrong answer (the second example).

## 4.5 DATASET TRANSFER

When jointly optimized, SmartRAG's policy network gains an awareness of both the companion knowledge base and its own capabilities. Specifically, the policy network knows what can or cannot be retrieved from the knowledge base. In addition, it is also aware of what itself knows or does not know. When transferring the optimized SmartRAG system to a previously unseen dataset, such kind of awareness can help the system achieve better performance. In this section, we test the model

| Method | TriviaQA | | 2WikiMultiHopQA | | MuSiQue | |
|---|---|---|---|---|---|---|
| | EM | F1 | EM | F1 | EM | F1 |
| SFT RAG | 47.72 | 56.99 | 23.08 | 28.21 | 3.27 | 11.15 |
| SKR (Wang et al., 2023) | 37.51 | 45.35 | 22.61 | 27.65 | 3.02 | 10.46 |
| SmartRAG | **48.87** | **59.17** | **23.55** | **29.26** | **4.84** | **13.33** |

Table 6: Dataset Transfer of Flan-T5-large.

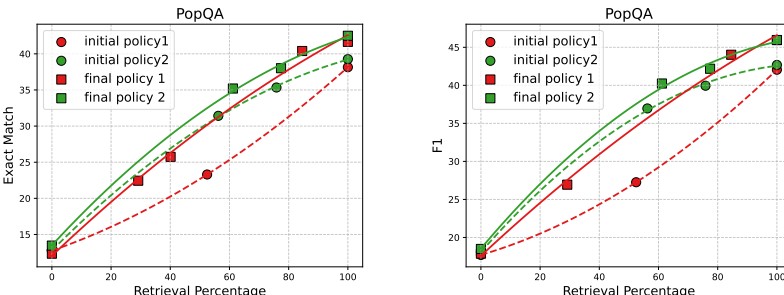

Figure 3: Influence of different initial policies for SmartRAG.

trained in Section 4.1 on three previously unseen datasets, namely *TrivialQA* (Joshi et al., 2017), *2WikiMultiHopQA* (Ho et al., 2020) and *MuSiQue* (Trivedi et al., 2022). We compare SmartRAG with two other baselines including **SFT RAG** and **SKR**(Wang et al., 2023). The result is shown in Table 6. SmartRAG achieves better transfer performance than both baselines, validating our argument.

To further demonstrate the awareness of both the knowledge base and the policy model itself, we categorize the TrivialQA test set into three types: 1) questions answerable without retrieval, 2) those requiring retrieval for a correct answer, and 3) those unanswerable even after retrieval. We fix the retrieving threshold to be $0.5$ and evaluate the retrieval ratio for each of the categories. Theoretically speaking, retrieval is only needed for the second type of data points. The model should not retrieve for the first type because it knows it already knows the answer. It will also not retrieve for the third type because it knows the knowledge base does not have such information. The retrieval ratio on all three types of data points for SmartRAG is $10.59\%$, $42.59\%$ and $29.04\%$ respectively, with the highest probability of retrieval for the second category, thereby affirming our assertion. Using different thresholds and alternative datasets may yield different results, but we always find that the second category has the highest retrieving ratio.

### 4.6 INFLUENCE OF THE INITIAL POLICY

We have proposed two initial policies in Section 3.1. To study how the initial policies can influence the final performance, we start from both initial policies and go through the same PPO procedure. The exact match and F1 score of the initial policy and final policy for both initializations are shown in Figure 3. $\pi_{\theta_0}$ (initial policy 1) clearly performs worse than $\pi_{\theta_0^*}$ (initial policy 2) because the corresponding warm up dataset contains less information. Applying PPO enhances both policies. The final policy with better initialization still outperforms its counterpart with worse initialization, indicating that a well-designed initial policy benefits the PPO procedure.

## 5 CONCLUSION

We introduce SmartRAG, a novel framework that can decide when to retrieve, what to retrieve, and how to answer with/without the retrieved observations. All these functions are accomplished by a policy network and a companion retriever. SmartRAG jointly optimizes the whole system using reinforcement learning. Empirical results show that SmartRAG outperforms previous methods whose modules are separately optimized. For systems comprising multiple modules, it is necessary to optimize them from a holistic perspective—a challenge we reserve for future research endeavors.

ETHICAL CONCERNS

This paper primarily focuses on an end-to-end RAG framework that addresses the questions of when to retrieve, what to retrieve, and how to answer within a single LLM. Our method utilizes open-source datasets and Bing retrieval for training and testing, with all data being publicly accessible. While our approach demonstrates the ability to effectively enhance the model's decision-making capabilities and improve the accuracy of retrieved information, as well as the final answers, we hope that this design—more aligned with real-world applications—can enhance the efficiency of RAG pipelines. Furthermore, we aspire for this promising framework to be applied across various LLM decision-making scenarios.

REPRODUCIBILITY STATEMENT

The details of experiment settings are provided in Section 4, and a more detailed description and implementation setting can be found in Appendix A. All datasets and models used in our work are publicly available. Additionally, we have uploaded our code in the supplementary materials and will open-source it upon acceptance of the paper.

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

APPENDIX

# A   SMARTRAG DETAILS

## A.1   DETAILS OF RAW DATASETS

A concise summary of the utilized datasets is presented below:

- PopQA (Mallen et al., 2023a) consists of 14k questions designed to encompass factual information in the long tail, potentially overlooked by widely used QA datasets. It is an entity-centric, open-domain QA dataset featuring entities with varying levels of popularity.

- AmbigNQ (Min et al., 2020) provides a disambiguated version of Natural Questions(NQ) (Kwiatkowski et al., 2019). For ambiguous questions in NQ, minimal constraints are introduced to refine them into multiple similar yet more specific questions

- HotpotQA (Yang et al., 2018) is a multi-hop dataset derived from Wikipedia. The questions are diverse and not limited by any predefined knowledge bases or schemas. HotpotQA is an English Wikipedia question-answering dataset consisting of approximately 113K crowd-sourced questions, designed to require the introductory paragraphs of two Wikipedia articles for answering. Each question is accompanied by two gold paragraphs and a set of sentences within these paragraphs, identified by crowd workers as supporting facts essential for answering the question.

- TriviaQA (Joshi et al., 2017) is a large-scale, text-based question-answering dataset consisting of 950K question-answer pairs derived from 662K documents, sourced from both Wikipedia and the web.

- MuSiQue (Trivedi et al., 2022) is constructed by combining multiple single-hop queries, where each reasoning step fundamentally depends on information from another. In our experiments, we simplified the task by treating it as a single-hop problem for testing purposes.

- 2WikiMultiHopQA (Ho et al., 2020) is a multihop QA dataset derived from Wikipedia. It requires two hops to answer a question. Similarly, in our paper, we treat it as a single-step test.

- OpenBookQA (Mihaylov et al., 2018) consists of 5,957 multiple-choice questions, each with four possible answers. The dataset is supplemented with external fundamental scientific facts, requiring a deep understanding of these facts to answer the questions accurately.

- MedQA-cn (Jin et al., 2021) compiles questions from the National Medical Board Examinations of Mainland China. It serves as a challenging benchmark, encompassing a wide range of medical knowledge, including patient profiles, disease symptoms, and drug dosage requirements. This diversity necessitates a contextual understanding to accurately respond to the questions presented.

- ARC-c (Clark et al., 2018) contains complex questions that demand deep reasoning, integration of multiple information sources, and advanced inferential capabilities. This subset pushes AI models beyond basic text matching, requiring them to address ambiguity and engage in sophisticated problem-solving, testing the depth of their reasoning and comprehension.

The scale of the data used in the paper is presented in Table 7. To reduce the number of training iterations, we combined the three datasets for joint training. It is worth noting that this cross-dataset training presents a more challenging task, and we employed multi-dataset training during both the warm-up and PPO training phases. To mitigate the long-tail issues arising from distribution differences among the datasets, we utilized only 20k samples from the HotpotQA dataset to align with the scales of the PopQA and AmbigNQ datasets. Additionally, for the multiple-choice questions, we limited our training to 2k samples from both OpenBookQA and MedQA-en.

## A.2   DETAILS OF WARM DATASET COLLECTION

In the earlier mentioned process of handling the warm-up dataset, we utilized datasets of the same scale as those used in training for multi-task warm-up. Each question presents three options: a direct response, generating a query, and responding based on the retrieved content. The answer content comes directly from the dataset, while the search query is generated by GPT-3.5. The prompt we

| Datasets | Category | Training Set | Test Set |
|---|---|---|---|
| PopQA | Open Domain | 13.0k | 0.7k |
| AmbigNQ | Open Domain | 19.4k | 5.8k |
| HotpotQA | Open Domain | 90.4k | 7.4k |
| TriviaQA | Open Domain | 78.8k | 8.8k |
| OpenBookQA | Scientific Knowledge | 5.0k | 0.5k |
| MedQA-en | Medical Domain | 10.2k | 1.3k |
| ARC-c | Commonsense reasoning | 1.1k | 1.2k |

Table 7: Details of raw datasets.

provided to GPT-3.5 is: 'Assuming you are a professional retrieval model, rewrite the following question into a query that is more suitable for retrieval: {Question}'. Based on this, the warm-up model responds to each question with a 50% probability of either providing a direct answer or an answer after retrieval, making it difficult to determine the importance of retrieval for each question. However, this setup offers reinforcement learning and a balanced initial state to explore different decision-making strategies.

As mentioned in Section 4.6, a better initial state can lead to improved results; therefore, we refined the SFT data for initial policy 2. We divided the data into two categories based on the model's ability to provide direct answers in the training set (F1 score $\geq 0.2$): one where the model can answer correctly to some extent, and another where the answers are entirely incorrect. During warm-up, for the first category, the model only provides direct responses, while for the second, it generates a query and answers based on the retrieved content. Under this warm-up training strategy, the model acquires a certain level of retrieval decision-making capability. This reasoning stems from the idea that if the model can provide correct answers on its own, there is no need to rely on external retrieval. Through joint training in our framework, the model further becomes aware of the effectiveness of the database, optimizes its query retrieval, and refines the answer generation, allowing it to explore a better state beyond the warm-up phase.

### A.3 TRAINING AND INFERENCE DETAILS

**Retriever**  We utilize the Bing search engine as our retriever, which does not require candidate index construction like dense retrievers, nor predefined candidates as in traditional resources. Instead, it offers a broad knowledge scope and ensures up-to-date factual information. We concatenate snippets from all retrieved web pages, selecting relevant sentences identified by Bing—much like entering a query into a browser, then gathering the text displayed on the search results page. The results obtained through this concatenation method resemble summaries of each webpage, encapsulating the core content of the retrieved information within a constrained length. The search engine will return multiple results given a query. We select the top $K$ results and concatenate the snippets from these results as the observation. In our experiments, we set $K = 4$ without other clarifications.

**Model**  Since our framework employs a single model to handle all tasks within the RAG system, we employed models of varying sizes for training. To further explore the effects of different architectures and model sizes, we conducted experiments on the Flan-T5 series models. In addition, we trained a Llama2 7B model to validate the effectiveness of our framework on a larger model. It is noteworthy that, since we use PPO for training, it requires optimizing the policy model, value model, and reference model simultaneously. To reduce memory consumption and improve training speed, we applied LoRA (Hu et al., 2021) for tuning LlaMa-7B.

**Settings**  We use 4 Nvidia A100 with 80GB memory to train our models. For the SFT warmup stage, we use the following training parameters: a learning rate of 3e-4, a batch size of 8, and to prevent overfitting, the number of epochs is set to 1. For reinforcement learning, we set the sampling steps to 5120, 10 threads, 512 steps for each. After sampling, the policy network is trained for 1 epoch, with a learning rate of 2e-6 and batch size of 32.

**Retrieval reward**  For the penalized retrieval reward $\alpha$ in equation 5, we assign a value of -0.2. This decision is motivated by the substantial influence this parameter exerts on the training dynamics. A high setting would drive the model to predominantly opt for retrieval, whereas a low setting would

discourage retrieval altogether. Given the explicit nature of this retrieval reward, any significant deviation from the retrieval-generated reward during the model's sampling process may prompt the model to adopt an overly simplistic strategy. Consequently, this could hinder the reinforcement learning process, limiting the model to either a retrieval-only or non-retrieval optimization approach, thus compromising its ability to sample diverse examples effectively.

---

**Direct Answer(Query) Instructions**
You will be presented with a question. If you know the answer, please respond directly. If you don't know the answer, use the Bing search engine to find the necessary information and then answer the question based on your observation.

Question: input

Please format your output as follows:

1. If you choose to answer the question directly, please use: '[Answer] YOUR_ANSWER'
2. If you choose to use the Bing search engine, please use:
'[Search] YOUR_SEARCH_QUERY'

Please output:

- - - - - - - - - - - - - - - - - - - - - - - - - - - - - - - - - - - - - - - - - - - - - -

**Retrieval Answer Instructions**
You will be presented with a question. If you know the answer, please respond directly. If you don't know the answer, use the Bing search engine to find the necessary information and then answer the question based on your observation.

Question: input

Observation: search

Please format your output as follows:

1. If you choose to answer the question directly, please use: '[Answer] YOUR_ANSWER'
2. If you choose to use the Bing search engine, please use:
'[Search] YOUR_SEARCH_QUERY'

Please output:

---

Table 8: Instructions for two-step RAG in SmartRetriverX.

**Generation Details** During the reinforcement learning training phase, to ensure sampling diversity and obtain samples reflecting different decision-making strategies, we employ Top-K decoding with K set to 50. This means that at each time step, the top K most probable candidate words from the probability distribution are selected, while the probabilities of the other words are set to zero. The probabilities of these K candidate words are then renormalized to sum to 1, forming a new probability distribution. A word is then randomly sampled from this new distribution as the output for the current time step. This process is repeated until an end token is generated or the maximum length is reached.

During inference, we use Beam Search for decoding. For the first token that determines when to retrieve, we apply a threshold-based approach (Asai et al., 2024): if the confidence score of the generated query exceeds the threshold, retrieval is performed; otherwise, the model directly provides an answer.

**List of Instructions** The entire inference process consists of either one or two steps. The first step involves either directly answering or generating a query. If a query is generated, a second step follows, which entails answering based on the content retrieved according to the query. Table

8 presents the instructions utilized in both steps. To ensure consistency in instruction during the fine-tuning process, we designed the two instructions to maintain as similar a format as possible; the instruction for the second step incorporates only one additional component: the retrieved content observation. We employed the same data format across all datasets, and for multiple-choice questions, we incorporated the candidate answers into the instruction of the question.

### A.4 FURTHER EXPLANATION ABOUT REWARD DESIGN

Next, we will illustrate through various examples how our reward design and training strategy enable SmartRAG to achieve optimal performance with minimal retrieval attempts.

Suppose a question that the LLM cannot corretly answer without retrieval, which will result in two distinct trajectories.

- Trajectory-1: [Answer](incorrect, reward=0)
- Trajectory-2: [Retrieve] (reward=-0.2) - [Answer] (correct, reward=2)

In the first trajectory, the value of the state [Answer] is since it gives the wrong answer. In the second trajectory, the value of the state [Retrieve] is $-0.2 + 2 * \gamma$, where $\gamma$ is the discount factor. In this case, the model will learn to use trajectory 2 to correctly answer the question since the value of the second trajectory is while that of the first trajectory is only .

When the LLM can directly answer the question without relying on retrieval, two different trajectories also emerge:

- Trajectory-1: [Answer] (correct, reward=2)
- Trajectory-2: [Retrieve] (reward=-0.2) - [Answer] (correct, reward=2)

In the first trajectory, the value of the state [Answer] is 2. In the second trajectory, the value of the state [Retrieve] is $-0.2 + 2 * \gamma (< 2)$. In this case, the model will learn to use trajectory 1 to answer the question. That being said, our algorithm not only learns how to correctly answer the question, but also learns how to use the minimal cost.

Therefore, the retrieval reward does not entirely discourage the model from performing retrieval. Instead, it facilitates a tradeoff between the benefits of retrieval and its associated costs, enabling the model to achieve higher answer accuracy with fewer retrieval attempts. Simultaneously optimizing the three tasks not only enhances retrieval efficiency but also improves the model's response quality, ultimately resulting in an LLM capable of addressing end-to-end RAG challenges effectively.

### A.5 INFERENCE LATENCY AND TRAINING TIME

Our inference pipeline is the same as many of the baselines such as GPT4 rewriter + SFT RAG and SKR. CoT-related algorithms like ReAct and IRCoT are much less efficient because they need extra computational cost for the intermediate thought and reflection. Considering that SmartRAG learns when to retrieve, it can automatically skip the retrieval step when unnecessary. As figure 3 shows, SmartRAG can achieve similar (or even better) performance than the baselines using less computational cost. In addition, table 2 demonstrates that SmartRAG refuses to retrieve when the database contains too few information such that it can save computational cost. As a result, our inference process achieves lower model costs and reduced retrieval costs compared to all other baselines.

In terms of training, our approach includes an additional RL training phase compared to standard SFT. While this introduces an extra training step, the overall improvement in inference performance and cost efficiency justifies this additional effort.

## B ANALYSIS

### B.1 TRAINING PROCESS

Our training process includes both warm-up training and reinforcement learning training. The figure illustrates the changes in iteration reward, KL reward, policy loss, and value loss during the rein-

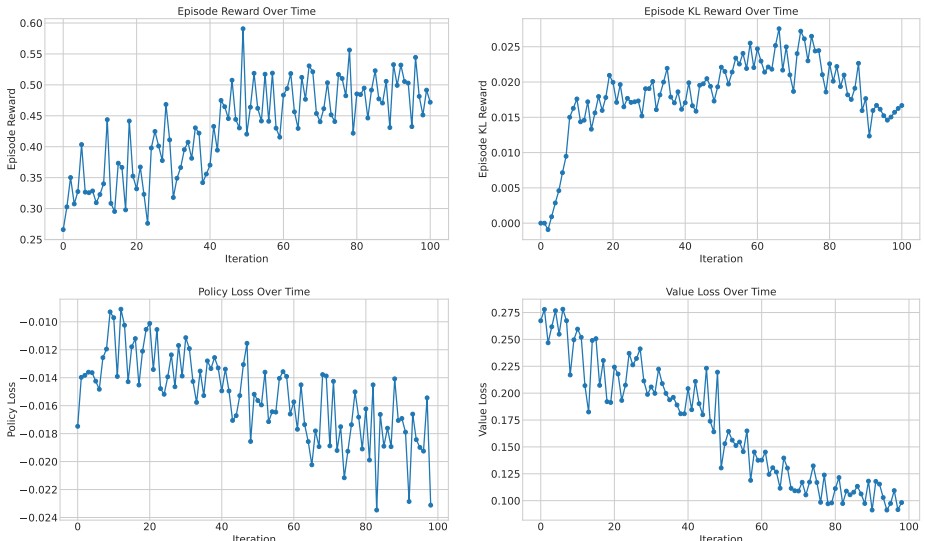

Figure 4: Iteration reward, KL reward, policy loss and value loss during training.

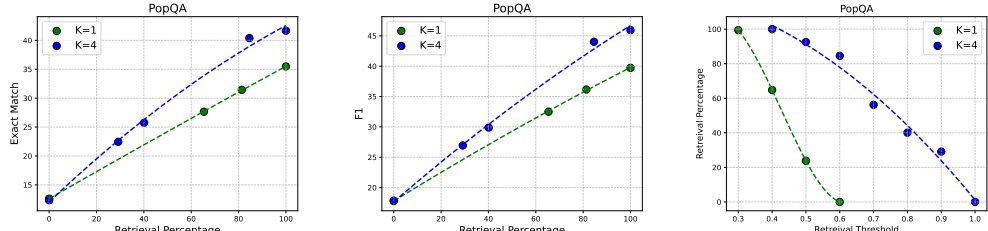

Figure 5: Exact Match, F1 Score, and Retrieval Threshold of different retrieval number K on Flan-T5-large.

forcement learning phase. The iteration reward, corresponding to the reward in the equation, shows an upward trend throughout the training process, indicating that the model is exploring in the right direction. The increase in KL divergence reward indicates that the model's strategy is being updated, with growing divergence from the previous strategy. At the same time, the loss trends of the policy model and value model are almost the opposite of the reward trend, which aligns with the typical training process of oscillating convergence followed by stable oscillations.

## B.2 RETRIEVAL NUMBER

In the retrieval-augmented framework, the number of retrieved texts also influences the model's performance. Generally, when the number of retrieved documents is not large, an increase in the number of retrieved documents leads to greater gains for the model. This is because when the retrieved information falls within the model's capacity to process, a larger set of documents is more likely to contain useful information. Therefore, we investigated the model's performance when the retrieval size is K=1 and K=4 based on initial policy 1. When K=4, it represents a retrieval of more documents and a stronger retrieval capacity, allowing us to explore the impact of varying-value database on our method's training results. The results, as shown in the figure, indicate that retrieving more documents leads to better performance. The figure also illustrates the relationship between the retrieval ratio and the threshold under different retrieval intensities. It is evident that when the retrieval database is more likely to contain relevant information, our method can demonstrate better awareness of the database, which in turn drives the model to rely more on retrieval. This aligns with real-world scenarios, where higher retrieval efficiency should increase the likelihood of the model

| Method | PopQA | | AmbigNQ | | HotpotQA | |
|---|---|---|---|---|---|---|
| | EM | F1 | EM | F1 | EM | F1 |
| LlaMa2-7B | 42.22 | 45.97 | 34.33 | 43.88 | 25.33 | 35.32 |
| GPT35 | 42.50 | 45.95 | 33.71 | 42.75 | 25.54 | 35.23 |

Table 9: Performance between different query generator on Flan-T5-large.

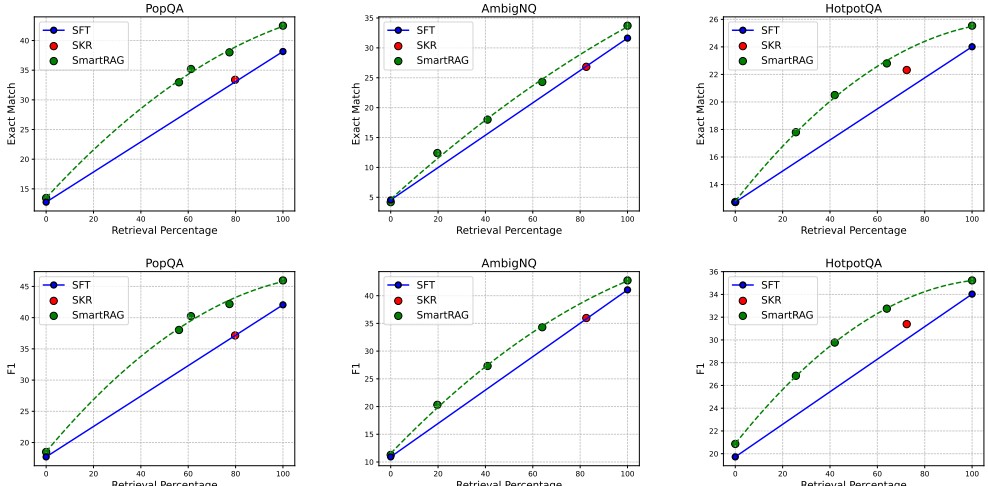

Figure 6: F1 Score of different retrieval percentage across three datasets on Flan-T5-large of initial policy 2.

engaging in retrieval, and vice versa. In contrast, the previous adaptive retrieval method was unable to recognize the impact of different databases.

### B.3    QUERY GENERATOR

To further evaluate the impact of different query generators on our results, we replaced GPT-3.5 with LLaMA-7B to generate the query dataset while keeping all other settings unchanged. We then trained our SmartRAG model on Flan-T5-large, and the results are presented in Table 1. It can be observed that the overall performance does not change significantly. This is because using a different LLM to generate the SFT query dataset is analogous to adopting a different initial policy for the RL step. Such variations in the initial state exert minimal influence on the final performance, provided the LLM is not exceptionally suboptimal. Moreover, the results from SFT + GPT-4 rewriter demonstrate that queries directly rewritten by GPT-4 are not optimal. This limitation arises because GPT-4 lacks prior exposure to the knowledge base. In contrast, the SmartRAG framework, which integrates a knowledge base into its training paradigm, enables query optimization that is better aligned with the target knowledge base.

### B.4    ADAPTIVE RETRIEVAL

Section 4.2 presents the F1 score results under the initial policy2 state. Here, we provide more comprehensive results with a different initial state, where the results for both initiate policy2 and initiate policy1 are shown in Tables 6 and 7, respectively. Nevertheless, we can observe that our approach enables the model to learn the concept of when to retrieve, resulting in improved final answers, which stem from the optimization of what to retrieve and how to answer.

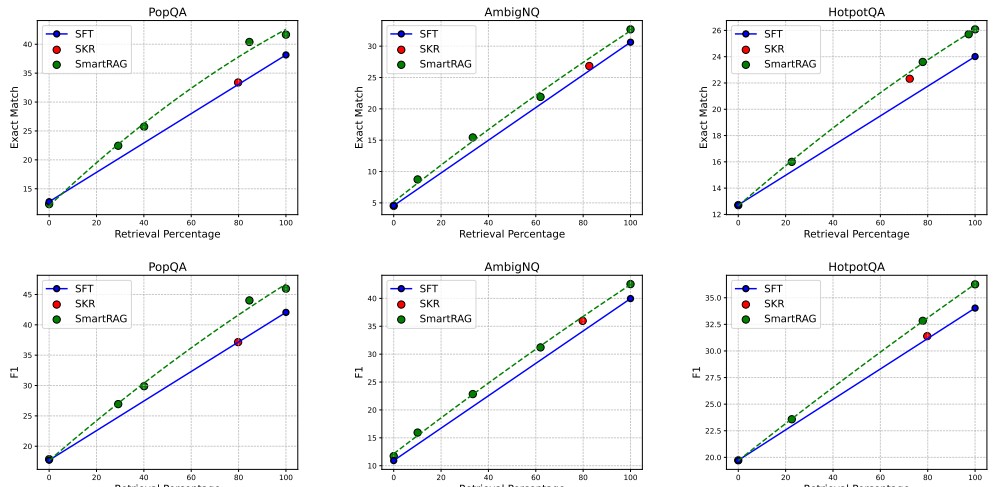

Figure 7: Exact Match and F1 Score of different retrieval percentage across three datasets on Flan-T5-large of initial policy 1.

### B.5 QUALITATIVE CASES

Sections 4.3 and 4.4 have already demonstrated that SmartRAG significantly improves both the 'what to retrieve' and 'how to answer' tasks through quantitative and qualitative analyses. Table 10 and 11 presents additional cases from three datasets. Similarly, it can be observed that the reformulated queries are more effective in retrieving relevant results, and under the same retrieval conditions, the model trained with PPO also demonstrates superior performance

**Question 1:** In what city was Rafael Reyes born?

**Golden Answer:** Cotija

**Observation with original question:** Rafael Reyes — Political leader, Conservative party — Britannica: Rafael Reyes (born 1850, Santa Rosa, New Granada2014died February 19, 1921, Bogot̆00e1, Colombia) was an explorer and statesman who was president and dictator of Colombia from 1904 to 1909. He attempted to give his nation a strong one-man rule that would attract foreign investment and foster domestic industrialization.

**Output:** [ANSWER] Santa Rosa

**Rewritten Query:** Birth city of Rafael Reyes

**Observation with rewritten query:** Rafael Reyes (Leafar Seyer) Age, Wife, Children, Family, Biography ...: Quick Info2192. Hometown: Cotija, Michoacan, Mexico. Wife: Kat Von D. Age: 47 Years. Some Lesser Known Facts About Rafael Reyes aka Leafar Seyer.

**Output:** [ANSWER] Cotija

**Question 2:** What other series is the actress who plays Malory Archer well known for?

**Golden Answer:** Arrested Development

**Observation with original question:** How 'Archer' said goodbye to Jessica Walter on its season finale: Walter, who voiced the character of spy agency chief Malory Archer on the animated comedy series, died in March at the age of 80. Deadline subsequently reported that Walter had been working...

**Output:** [ANSWER] The Simpsons

**Rewritten Query:** Which other series is the actress known for playing Malory Archer known for?

**Observation with rewritten query:** Jessica Walter — Archer Wiki — Fandom: She performed the voice of Malory Archer, who was strongly influenced by Walter's aforementioned role as Lucille Bluth on Arrested Development, in Archer for the first 12 seasons. Walter passed away on March 24th 2021. Season 12 served as her final television role and was released following her death. Its final episode was dedicated to her ...

**Output:** [ANSWER] Arrested Development

**Question 3:** Wheelock Whitney is just one member of the Whitney Family. Where did this American family originate from?

**Golden Answer:** London, England

**Observation with original question:** Minnesota businessman Wheelock Whitney Jr. dies — AP News: Wheelock Whitney, whose fraternity brothers at Yale University included former President George H.W. Bush, was born to a prominent family in St. Cloud. He was CEO from 1963-1972 of the investment banking firm Dain and Co., which he helped turn into a regional financial powerhouse and eventually became RBC Wealth Management.

**Output:** [ANSWER] St. Cloud

**Rewritten Query:** What is the origin of the Whitney Family, including Wheelock Whitney?

**Observation with rewritten query:** Whitney History, Family Crest and Coats of Arms: Origins Available: England. Ireland. Whitney is one of the many names that the Normans brought with them when they conquered England in 1066. The Whitney family lived in Herefordshire, at the village of Whitney. Early Origins of the Whitney family.

**Output:** [ANSWER] England

Table 10: Comparison between observations with orignal questions and rewritten queries.

**Question 1:** In what country is Dell?

**Golden Answer:** America

**Observation:** Where Are Dell Computers Made?: Dell 2018s Current Manufacturing Footprint and Supplier Ecosystem. Flash forward to 2023, and Dell now operates major PC manufacturing plants in the Chinese cities of Xiamen, Chengdu, Wuhan, and Shenzhen. Dell also owns enormous PC production facilities just over the border from mainland China in Taipei and Taoyuan, Taiwan... The other Dell assembly plants are located in China, Malaysia, Brazil, and the United States. Here's Where Dell Computers And Laptops Are Made Today: Aside from the United States, Dell's ODM and final assembly facilities are in China, Vietnam, Taiwan, Mexico, Brazil, Ireland, Malaysia, India, Poland, and Thailand. Where Are Dell Computers Assembled? - DeviceTests: Mexico: Dell outsources the assembly of its computers to Foxconn in Mexico. Malaysia: Dell2019s assembly factory is located in Penang, Malaysia. Other places where Dell computers are assembled include Ireland, the United States of America, Singapore, Vietnam, Japan, etc.

**Before PPO Output:** [ANSWER] Canada

**After PPO Output:** [ANSWER] America

**Question 2:** What is the religion of Paul?

**Golden Answer:** Christianity

**Observation:** Paul the Apostle World History Encyclopedia: A Founder of Christianity. In the last century, scholars have come to appreciate Paul as the actual founder of the religious movement that would become Christianity. Paul was a Diaspora Jew, a member of the party of the Pharisees, who experienced a revelation of the resurrected Jesus. Paul and Jesus, The Origin of Paul's Religion, John Gresham Machen ...

**Before PPO Output:** [ANSWER] Anglicanism

**After PPO Output:** [ANSWER] Christianity

**Question 3:** Where was the think tank founded by a neoconservative political analyst born in 1952 established?

**Gold Answer:** Washington, D.C.

**Search Query:** Establishment location of the think tank founded by a 1952-born neoconservative political analyst

**Observation:** Think tanks and the knowledgež2013policy nexus in China: For decades, institutionalized linkages with the party-state have shaped intellectual activity and subsequently, think tank agency; think tanks were anointed as either establishment oriented, being officially yoked to the party apparatus, or semi-establishment, operating with a modicum of independence but with the consent of governmental patrons. Think tanks and the knowledgež2013policy nexus in China: For decades, institutionalized linkages with the party-state have shaped intellectual activity and subsequently, think tank agency; think tanks were anointed as either establishment oriented, being officially yoked to the party apparatus, or semi-establishment, operating with a modicum of independence but with the consent of governmental patrons.

**Before PPO Output:** [ANSWER] New York City

**After PPO Output:** [ANSWER] Washington, D.C.

Table 11: Improvement of answer from observation between before and after PPO.

