# OpenReview forum: "SmartRAG: Jointly Learn RAG-Related Tasks From the Environment Feedback"
_ICLR.cc/2025/Conference — ICLR 2025 Poster_

### Official Review · Reviewer_FNgY · 2024-10-19

**Soundness:** 3
**Presentation:** 3
**Contribution:** 2
**Rating:** 6
**Confidence:** 4

**Summary:**

The paper proposes a jointly optimization method called SmartRAG which combines the retriever, the query rewriter and the answer generator. The authors mainly use a policy network which is powered by a LLM to decide whether to retrieve or answer directly. They set up a quota limitation, the policy network can only answer the query when the quota is reached. A supervised warm-up with constructed dataset which contains direct answer, retrieved answer and observed answer is firstly applied to the policy network. This step ensures the policy network with the ability to output a [retrieve] or [answer] action. Then they design a reward at time step t of either answer or retrieve and use PPO to optimize the policy network. SmartRAG can decide the action and generate answers with/without retrieved observations.

**Strengths:**

1. I like the idea of this paper, it consider a real problem LLM applications where the pre-trained language models and downstream RAG applications are built and optimized separately.
2. The paper is well-written, most ideas are clearly illustrated.
3. SmartRAG uses reinforcement learning methods to optimize its policy network, this is reasonable since states, actions and rewards observed in previous time steps can be used as environmental feedback instead of human feedback.

**Weaknesses:**

1. In the methodology part, the authors should address why there is a pre-specified quota for retrieval. RAG technicals are designed to address the lack of specific domain knowledge in the training data of a pre-trained LLM. It doesn’t make sense for the LLM-driven policy network being forced to generate the [answer] action instead of [retrieve] action when the retrieve times met the quota. I’ve never seen such a setting in industrial RAG applications. Retrieval seems necessary and provides solid answers, thus shouldn’t cause penalty even the retrieval cost of similarity search in a knowledge base do exist.
2. The optimization, however, seems not fully end-to-end. The knowledge base, which serves as a fundamental role in a RAG system, is ignored in the the design (Algorithm1 SmartRAG pipeline) and rarely mentioned in the paper. If the knowledge base is involved in the process, in methodology part, the optimization should not be tailored for specific query datasets, but remains effective even for unseen queries.

**Questions:**

1. In equation (1) where os is the concatenation of all the historical observations returned from the retriever, how do you solve the problem of LLM token limit?
2. Why [retrieve] action suffers from a 100% penalty in equation (5) of the reward design? If retrieval cost contributes to a negative reward, how about the correct answer generated from retrieved observations? Will that lead to an increase in the reward?

---

> ### Author Response · Authors · 2024-11-18
> **Respond to reviewer FNgY**
>
> We thank the reviewer for pointing out some unclear descriptions in our paper. We will address the questions below. Further discussions are welcomed if there are still misclarifications.
>
> **1. Why there is a pre-specified quota for retrieval？**
>
> We agree that a solid answer is more critical than retrieval cost in some scenarios. However, there do exist cases when retrival/time cost matters. Here are two examples:
>
> 1) Suppose we are doing a QA task, it is possible that the model will keep retrieving the database as long as the model thinks that the correct answer has not been retrieved, making the whole system stuck and not producing the final answer. Setting a pre-specified quota for retrieval helps avoid this problem.
>
> 2) For a chatbot, when the user talks about some super fresh news, the bot is expected to use the search engine to retrieve related informations, which of course will take additional time. If the user waits for too long, he/she might be impatient and leave the chat room.
>
> Of course **the pre-specified quota is NOT a mandatory design**. We can simply remove it if it is not necessary in a specific application.
>
> In our paper, we set the retieval quota to 1 to ensure a fair comparison with existing methods. Current methods such as SFT-RAG and other approaches typically perform only a single retrieval step.
>
> **2. The optimization seems not fully end-to-end for not including the knowledge base.**
>
> The knowledge base in our framework is treated more as a utility tool. So it is not part of the optimization target. However, we do consider the knowledge base in the optimization by learning how to better utilize the tool. That is, our LLM learns how to write a good search query for the knowledge base. The results in Table 4 demonstate that our result can significantly enhance the accuracy of retrieval matching.
>
> **3. How to solve the problem of LLM token limit?**
>
> In our practice, the concatenated of historical observations does not encounter the LLM token limit issue.
>
> We argue that all the LLM applications, including RAG, may encounter the token limit issue. Techniques like long-context LLMs may be a research direction to systematically solve the problem. But this is beyond the scope of this paper.
>
> **4. Why [retrieve] action suffers from a 100% penalty in equation (5) of the reward design?  how about the correct answer generated from retrieved observations?**
>
> We can understand the design by two examples. Suppose a question that the LLM cannot corretly answer without retrieval. Think about the following two trajectories:
>
> **Trajectory-1**: [Answer]（incorrect, reward=$0$）
>
> **Trajectory-2**: [Retrieve] (reward=$-0.2$) - [Answer] (correct, reward=$2$)
>
> In the first trajectory, the value of the state [Answer] is $0$ since it gives the wrong answer. In the second trajectory, the value of the state [Retrieve] is $-0.2 + \gamma * 2$, where $\gamma$ is the discount factor ($0.9$). In this case, the model will learn to use trajectory 2 to correctly answer the question since the value of the second trajectory is $1.6$ while that of the first trajectory is only $0$.
>
> Now let us think about another example, which the LLM can correctly answer without any retrieval. Again we have two trajectories:
>
> **Trajectory-1**: [Answer] (correct, reward=$2$)
>
> **Trajectory-2**: [Retrieve] (reward=$-0.2$) - [Answer] (correct, reward=$2$)
>
> In the first trajectory, the value of the state [Answer] is $2$. In the second trajectory, the value of the state [Retrieve] is $-0.2 + \gamma * 2 ~~~ (<2)$. In this case, the model will learn to use trajectory 1 to answer the question. That being said, our algorithm not only learns how to correctly answer the question, but also learns how to use the minimal cost.
>
> Again **the retrieval cost is NOT mandatory**, we can remove it if retrieving cost is not an issue in a specific application. If so, trajectory-1 and trajectory-2 will be equally good in the second example.

---

> > ### Comment · Reviewer_FNgY · 2024-11-26
> >
> > Thank you for your response. You have addressed some of my concerns regarding the reward design, and I have updated my scores accordingly. However, I still have questions about the design. I think RAG is typically used without requiring fine-tuning for a specific dataset. In contrast, your design seems to necessitate fine-tuning the model for each individual dataset. This is why I mentioned "The optimization seems not fully end-to-end for not including the knowledge base" in my previous review.

---

> > > ### Author Response · Authors · 2024-11-27
> > >
> > > We thank the reviewer for the response and further discussion. We are sorry for our misunderstanding about the reviewer's original question.
> > >
> > > Actually SmartRAG learns the properties of the knowledge base, meaning that
> > >
> > > 1) it knows what query to use for unseen queries/datasets
> > > 2) it knows what information can or cannot be retrieved from the database
> > >
> > > We validate both arguments in **Appendix B.3 Dataset Transfer**. We explain a bit more here. We directly transfer our trained model in Section 4 to a **previously unseen dataset** TrivialQA and study the performance.
> > >
> > > We first evaluate the EM and F1 score on TrivialQA dataset and compare our algorithm with two baselines: SFT RAG and SKR. As shown in table 8 (copy and pasted below), SmartRAG achieves better transfer performance on both metrics.
> > >
> > > | Method   | EM | F1 |
> > > | -------- | --------- | --------- |
> > > | SFT RAG | 47.72 | 56.99 |
> > > | SKR | 37.51 | 45.35 |
> > > |SmartRAG | **48.87** | **59.17** |
> > >
> > > We further categorize TrivialQA into three types
> > > 1) questions that can be correctly answered without retrieval
> > > 2) questions that can only be correctly answered after retrieval
> > > 3) questions that cannot be correctly answered even after retrieval
> > >
> > > Theoretically speaking, a good policy should only retrieve on the second type of data points. By adjusting the retrieving threshold, SmartRAG is more likely to retrieve indeed on the second type. Here is an example retrieving ratio on all three types: **10.59, 42.59, 29.04**. For different thresholds, the retrieving ratios will be different but overall the retrieving ratio on the second type is the highest, indicating that SmartRAG is aware of what can and cannot be retrieved from the knowledge base.
> > >
> > > We will revise the paper and make this point more clear.

---

> > > ### Author Response · Authors · 2024-11-28
> > > **Paper revision about dataset transfer performance**
> > >
> > > We further revised our paper by removing the dataset transfer section in the appendix to the main paper, showcasing that our algorithm outperforms two baselines when transferring to new datasets. Two additional datasets are adopted and the results are also pasted below
> > >
> > > | Method   | 2WikiMultiHopQA(EM) | 2WikiMultiHopQA(F1) | MuSiQu(EM) | MuSiQu(F1) |
> > > | -------- | --------- | --------- | --------- | --------- |
> > > | SFT RAG | 23.08 | 28.21 | 3.27 | 11.15 |
> > > | SKR | 22.61 | 27.65 | 3.02 | 10.46 |
> > > |SmartRAG | **23.55** | **29.26** | **4.84** | **13.33** |
> > >
> > > Table 8 in the original paper becomes table 6 in our revised version.

---

> ### Author Response · Authors · 2024-11-21
> **Paper revision**
>
> To avoid misunderstandings, we revised the paper as follows:
>
> - We explicitly emphasized that the retrieval quota (line 155) and the retrieval penalty (line 219) are not mandatory in the revised paper.
>
> - We add a new subsection (section A.4) to further explain the reward design. By two examples, we demonstrate that such a design encourages the best performance with the lowest retrieval cost.

---

### Official Review · Reviewer_a9rm · 2024-11-03

**Soundness:** 2
**Presentation:** 2
**Contribution:** 2
**Rating:** 5
**Confidence:** 3

**Summary:**

This paper proposes an integrated framework that includes multiple components of RAG. The intuition is that jointly training these components can yield optimal performance. The experiments are conducted on four benchmarks with several baselines and two backbone models. The author also conducts ablation studies to investigate SmartRAG's behavior in terms of when to retrieve, what to retrieve, and how to answer.

**Strengths:**

1. The proposed SmartRAG is clear and easy to understand. Some of the designs are intuitive.
2. The paper writing is relatively smooth and easy to follow.

**Weaknesses:**

1. Part of the setups of SmartRAG are not well-justified. Please take a look at the questions.
2. Certain experimental observations are not elaborated clearly. Please take a look at the questions.
3. The RAG baselines seem not extensive, compared with references on this topic.

**Questions:**

1. Why does the policy network serve three roles in the model? What's the motivation behind it? Won't these three roles conflict with each other?
2. The optimization of the whole system is dependent on the feedback from the RL framework. How to distinguish whether a low reward is caused by a wrong policy or inaccurate retrieval?
3. Why is the retrieval percentage of SKR in Figure 2 fixed at 0.8?
4. In Section 4.1, the author states that *OpenBookQA, MedQA-cn, and ARC-c datasets* have nothing to retrieve from the Bing search engine. If that's the case, why does SFT RAG yield the best performance in 2 out of 3 cases with the highest retrieval ratio?

---

> ### Author Response · Authors · 2024-11-18
> **Respond to reviewer a9rm**
>
> We thank the reviewer for the detailed questions and we are sorry for some confusions in the original paper. A better clarification will definitely improve the paper and help the readers better understand it.
>
> **1. Why does the policy network serve three roles in the model? What's the motivation behind it? Won't these three roles conflict with each other?**
>
> The policy network is implemented by an LLM. Thanks to the multitask capability of LLMs, our policy network can serve three different roles. This is an advantage comparing to methods that need different networks in the sense of saving deployment model size.
>
> Theoretically speaking, there may be conflicts between different tasks, especially when the network capacity is not high enough. However, in practice, this is not an issue for morden LLMs. Our empirical results demonstrate that SmartRAG achieves better performance than the baselines.
>
> In addition, our algorithm does not require to use a single model for all the three tasks. We can use different models and optimize the whole system using exactly the same training procedure.
>
> **2. How to distinguish whether a low reward is caused by a wrong policy or inaccurate retrieval?**
>
> We regard the search engine/knowledge base as an external tool. To accurately retrieve the related information, our policy network should produce a good query for retrieval. In this sense, inaccurate retrieval is a form of poor policy, which will be optimized using the RL pipeline. The results in Table 4 demonstate that our result can significantly enhance the accuracy of retrieval matching, by learning how to better utilize the tool.
>
> **3. Why is the retrieval percentage of SKR in Figure 2 fixed at 0.8?**
>
> SKR is based on a classifier and produces a deterministic result being either retrieval or no retrieval. In contrast, SmartRAG adjusts the retrieval confidence threshold, allowing for a range of retrieval percentages, rather than a fixed decision.
>
> **4. Why does SFT RAG yield the best performance in 2 out of 3 cases with the highest retrieval ratio? Which is conflcit to the statements.**
>
> SFT RAG yiled **marginally higher accuracy** in table 2 with **MUCH higher retrieval cost** comparing to SmartRAG. It should be pointed out that our goal is to achieve **the highest accuracy with the LOWEST retrieval percentage**. Sorry for the confusion. We will better clarify this in table 2, mentioning that higher accuracy and lower retrieval percentage is better.
>
> In table 2, SFT RAG only outperforms SFT by about one point on average, suggesting that RAG fails to retrieve relevant content on these datasets in approximately 99% of cases. SKR cannot learn this property and still tries to retrieve in more than 84% questions on MdeQA-cn dataset. In contrast, SmartRAG learns the property of the knowledge base and choose not to retrieve to save retrieving cost.
>
> **5. The RAG baselines seem not extensive, compared with references on this topic.**
>
> We compared with 7 baselines from 3 different categories in table 1. We do skip some baselines like SKR and SelfRAG in table 1 for fair comparison. However, we include SKR in figure 2 and table 2, where a more fair setting is adopted.
>
> We thank the reviewer for pointing this out and we will adding more baselines to table 1 to enrich the results. This will definitely improve the quality of our paper. Some of the baselines are demonstrated below. If the reviewer has more suggestions about the baselines, we are happy to include the results of these methods.
>
> | Method   | PopQA(EM) | PopQA(F1) | AmbigNQ(EM) | AmbigNQ(F1) | HotpotQA(EM) | HotpotQA(F1) |
> | -------- | --------- | --------- | ----------- | ----------- | ------------ | ------------ |
> | SelfRAG | 1.07      | 23.49     | 6.12        | 26.48       | 6.80         | 17.53        |
> | SKR      | 36.04     | 39.59     | 35.41       | 44.75       | 23.63        | 33.44        |
> | SmartRAG | 44.32     | 47.60     | 37.89       | 47.76       | 26.60        | 36.42        |

---

> ### Author Response · Authors · 2024-11-21
> **Paper revision**
>
> As the reviewer suggests, we have made the following modifications to the paper:
>
> - Three more baselines are added to table 1, including *SKR*, *GPT4 rewriter + SKR* and *Self-RAG*. Two baselines are added to figure 2.
>
> - Table2 is updated to clearly show that lower retrieval percentage is better.

---

### Official Review · Reviewer_47ez · 2024-11-03

**Soundness:** 3
**Presentation:** 3
**Contribution:** 2
**Rating:** 8
**Confidence:** 3

**Summary:**

This paper proposes a new Smart RAG framework that consists of two parts. (1)Policy Network and (2) Retriever. The policy network decides whether to retrieve or not based on the input state (question + initial observation), and generates an appropriate query to obtain the required information form the retriever. The system is trained in an end-end method using supervised fine tuning and PPO.

**Strengths:**

1. The main contribution of this work is the end to end trining of the RAG framework. The authors do a good job in analyzing the proposed framework (Section 4.1-4.4)

2. The paper is generally well written.

**Weaknesses:**

The proposed method is not generalizable.

1. The proposed method is very domain specific and requires the training of the entire module for each dataset.
2. The generalizability of the approach is further curtailed by the availability of datasets with true final answers. (Which is needed for SFT and  PPO)
3. The end to end framework requires complete retraining if any part of the framework is changed.

**Questions:**

1. Clarification on the training dataset.
    1. Was all of the training data used for SFT and PPO?
    2. On which dataset was the EM and F1 evaluations done? Was this used as a part of the SFT or PPO process?
2. Is SFT RAG the performance of $\pi_{\theta^*_0}$ in the setting described in Figure 1? (Was SFT RAG evaluated in the same setting as SmartRAG? where the policy network was allowed one retrieval? )
3. Assuming the availability of a powerful model (GPT-3.5) for generating the dataset for SFT is not always true. Have the authors tried using the base LLM to generate the SFT dataset?

---

> ### Author Response · Authors · 2024-11-18
> **Respond to reviewer 47ez**
>
> **1. The proposed method is very domain specific and requires the training of the entire module for each dataset.**
>
> We argue that any finetuning is specific to a particular domain. Any SFT of LLM requires the training of the entire model for each dataset. Parameter efficient finetuning can also be applied to enhance the training efficiency of our SmartRAG. In our experiments, we finetune the whole model for Flan-T5 and use LoRA to finetune Llama.
>
> **2. The generalizability of the approach is further curtailed by the availability of datasets with true final answers. (Which is needed for SFT and PPO)**
>
> Firstly, most finetuning rely on the golden output (or preference pair, which includes kind of golden answer).
>
> Secondly, our SmartRAG actually requires less golden supervision. We do not require the golden answer for *when to retrieve* and *what to retrieve* but only the final answer of the question. Other methods even need more golden outputs. For example, Self-RAG depends on supervisory signals generated by GPT-4 to determine whether a question requires retrieval.
>
> Lastly, our algorithm does not necessarily depend on the final answer. As long as we can design a kind of reward, we can get rid of the direct supervision. This is one of our key ideas to remove the supervision of the intermediate modules, which we have discussed from line 37-40 in our submission.
>
> **3. The end to end framework requires complete retraining if any part of the framework is changed.**
>
> Theoretically speaking, changing any part of any system requires the other parts to adapt to the change. **The need for retraining arises not from the end-to-end design, but is necessitated by the alignment of the training and test distributions.** Thanks to the great generalization ability of LLMs, the performance degration of LLM systems when transferred to other settings is usually within tolerance in many situations. Of course further finetuning can enhance the performance.
>
> Our SmartRAG inherits the good generalization property of LLMs. No retraining is needed with moderate setting descrepancy. For example, in section B.3, we validate that our pipeline can transfer to a new dataset without any adaptation.
>
> **4. Was all of the training data used for SFT and PPO?**
>
> Yes, we used the same training data for both the SFT and PPO processes. The training data is introduced in the beginning of section 4, with detailed data description in Appendix A1.
>
> **5. On which dataset was the EM and F1 evaluations done? Was this used as a part of the SFT or PPO process?**
>
> For all the datasets, we train on the training split and evaluate on the test split. Following standard procedure, **the training split and the test split has NO overlap**.
>
> **6. Was SFT RAG evaluated in the same setting as SmartRAG? where the policy network was allowed one retrieval?**
>
> Yes, SFT RAG is evaluated in the same way as SmartRAG in Figure 1. The network is allowed for one time of retrieval.
>
> **7. Have the authors tried using the weaker LLM to generate the SFT query dataset?**
>
> Using a different LLM to generate the SFT query dataset is like using a different initial policy for the RL step. Such a different inital state will not have much impact on the final performance if the LLM is not extremely bad. We tried replacing GPT-3.5 with LlaMA2-7B to generate our dataset but the performance is not much different. So we did not include the result in our paper. The results on Flan-T5-large are as follows.
>
> | Query Generator | PopQA(EM) | PopQA(F1) | AmbigNQ(EM) | AmbigNQ(F1) | HotpotQA(EM) | HotpotQA(F1) |
> | --------------- | --------- | --------- | ----------- | ----------- | ------------ | ------------ |
> | LlaMa2-7B       | 42.22     | 45.97     | 34.33       | 43.88       | 25.33        | 35.32        |
> | GPT35           | 42.50     | 45.95     | 33.71       | 42.75       | 25.54        | 35.23        |
>
> We can add this result in our revised paper per the reviewer's suggestion.

---

> > ### Comment · Reviewer_47ez · 2024-11-19
> >
> > Thank you for your response, I have updated my score accordingly. All the best!

---

> > > ### Author Response · Authors · 2024-11-21
> > >
> > > We sincerely thank the reviewer for raising the score! All the best!

---

### Official Review · Reviewer_mFWs · 2024-11-04

**Soundness:** 2
**Presentation:** 3
**Contribution:** 2
**Rating:** 5
**Confidence:** 3

**Summary:**

The paper introduces SmartRAG, a retrieval-augmented generation (RAG) framework designed to enhance performance through joint optimization of its modules. Unlike traditional RAG approaches, SmartRAG uses a policy network that dynamically decides when to retrieve information, what queries to use, and how to synthesize answers based on retrieved observations. Reinforcement learning with environment feedback is employed to jointly train the system, improving coordination between modules and reducing retrieval costs. Experimental results on multiple QA datasets show that SmartRAG outperforms separately optimized systems.

**Strengths:**

a) The paper introduces reinforcement learning to jointly optimize RAG components, integrating retrieval decisions, query rewriting, and answer generation as policy-driven actions, representing progress in RAG architectures.

b) The introduction of reinforcement learning further demonstrates the flexibility of RAG and explores the potential of integrating RAG with other AI technologies.

c) Evaluation across several datasets demonstrates the effectiveness of the approach.

**Weaknesses:**

a) There exists many works focused on the question of "When to retrieval" in RAG, but in Section 4.1, only one baseline was selected for comparison. Incorporating a set of baseline examples into these studies would enhance comparative and analytical insights.

b) The current baselines primarily apply process-wide optimizations, such as fine-tuning the generator, without optimizing individual modules (e.g., rewriter, generator, and decision-maker) separately. This setup limits the ability to demonstrate how joint optimization in SmartRAG resolves potential conflicts and enhances coordination across modules.

Detailed comments:
a) A dedicated section on the limitations and assumptions underlying SmartRAG, particularly regarding the potential latency issues in real-world applications, would be beneficial.

b) While effective, the joint optimization approach may introduce complexity that could limit the model’s scalability, especially when applied to very large or real-time systems. Additional discussion on computational overhead and training time would strengthen the paper.

c) Considering the potential for optimizing PPO's efficiency or exploring alternative RL algorithms to reduce training costs, it could be beneficial to further discuss these aspects, which may enhance the practicality of the framework.

**Questions:**

a) There exists many works focused on the question of "When to retrieval" in RAG, but in Section 4.1, only one baseline was selected for comparison. Incorporating a set of baseline examples into these studies would enhance comparative and analytical insights.

b) The current baselines primarily apply process-wide optimizations, such as fine-tuning the generator, without optimizing individual modules (e.g., rewriter, generator, and decision-maker) separately. This setup limits the ability to demonstrate how joint optimization in SmartRAG resolves potential conflicts and enhances coordination across modules.

Detailed comments:
a) A dedicated section on the limitations and assumptions underlying SmartRAG, particularly regarding the potential latency issues in real-world applications, would be beneficial.

b) While effective, the joint optimization approach may introduce complexity that could limit the model’s scalability, especially when applied to very large or real-time systems. Additional discussion on computational overhead and training time would strengthen the paper.

c) Considering the potential for optimizing PPO's efficiency or exploring alternative RL algorithms to reduce training costs, it could be beneficial to further discuss these aspects, which may enhance the practicality of the framework.

---

> ### Author Response · Authors · 2024-11-19
> **Respond to reviewer mFWs**
>
> Thank you so much for your insightful questions and suggestions. These suggestions will definitely enhance our paper. We will discuss each of the reviewer's point and revise our paper accordingly.
>
> **1. In Section 4.1, only one baseline was selected for comparison.**
>
> We thank the reviewer for this suggestion. We will add more baselines including *SelfRAG* and *SKR + GPT4 Rewriter* for the comparison in section 4.1. The results are as follows. The number outside paranthesis is the accuracy (**higher is better**) while that in parentheses indicates the retrieval percentage (**lower is better**).
>
>
> | Method              | PopQA(EM)   | PopQA(F1)   | AmbigNQ(EM) | AmbigNQ(F1)  | HotpotQA(EM) | HotpotQA(F1) |
> | ------------------- | ----------- | ----------- | ----------- | ------------ | ------------ | ------------ |
> | SelfRAG             | 1.07(92.3)  | 23.49(92.3) | 6.12(97.3)  | 26.48(97.3)  | 6.80(100)    | 17.53(100)   |
> | SKR                 | 33.38(79.8) | 37.15(79.8) | 26.83(82.6) | 35.97 (82.6) | 22.32(72.4)  | 31.39(72.4)  |
> | SKR + GPT4 Rewriter | 34.23(79.8) | 37.76(79.8) | **28.37**(82.6) | 37.02 (82.6) | 22.39(72.4)  | 31.59(72.4)  |
> | SmartRAG            | **38.01**(**77.4**) | **42.18**(**77.4**) | 27.91(**68.9**) | **37.23**(**68.9**)  | **22.8**(**64**)     | **32.75**(**64**)    |
>
> We would also explain why we only include SKR as the baseline in section 4.1. Existing research on adaptive retrieval (when to retrieve) can be broadly categorized into two types. The first type is classifier-based methods (e.g., SKR), while the second type leverages GPT-4-generated data to guide the learning of this task (e.g., Self-RAG). The reason that we only include the first category is that its setting is the same as SmartRAG. This can make the comparison more fair.
>
> We agree with the reviewer that adding more baselines will certainly enhance the paper. We will include the above results in the revision per the reviewer's suggestion. If the reviewer suggests any other baseline, we will also be happy to include it.
>
>
> **2. The current baselines primarily apply process-wide optimizations, without optimizing individual modules separately.**
>
> Comparing with baselines that separately optimize individual modules is extremely important for our paper. We compare different combinations of separate training and joint training in different parts. We are sorry for the confused demonstration. However, We do demonstrate the advantages of our approach over separate training at different places. As the reviewer suggests, we will revise the paper and make this point more clear. Before this, we would like to clarify where we include these results in the original paper.
>
> **SFT RAG** in table 1 can be regarded as only training the answer generator.
>
> **GPT4 Rewriter + SFT RAG** in table 1 separately trains the answer generator and the query rewriter and combines them together at the inference time.
>
> **SKR** in figure 2 and table 2 separately trains the decision maker and the answer generator.
>
> **SKR + GPT4 Rewriter** separately trains all three modules and the performance is worse on most of the metrics than SmartRAG.
>
> **Replace Query** in table 4 only jointly trains the decision maker and the answer generator and thus the performance is worse than jointly training all three modules.
>
> **Replace Generator** in table 4 only jointly trains the decision maker and the query rewriter and again the performance becomes worse.

---

> ### Author Response · Authors · 2024-11-19
> **Respond to reviewer mFWs continue**
>
> **3. Discussion on the potential latency, computational cost and training time will be benefical.**
>
> We thank the reviewer for pointing the computation aspect out. Explicitly discussing this will definitely make the paper better. Our inference pipeline is the same as many of the baselines such as *GPT4 rewriter + SFT RAG* and *SKR*. CoT-related algorithms like *ReAct* and *IRCoT* are much less efficient because they need extra computational cost for the intermediate thought and reflection. Considering that SmartRAG learns when to retrieve, it can automatically skip the retrieval step when unnecessary. As figure 3 (and also the table in this response) shows, SmartRAG can **achieve similar (or even better) performance** than the baselines **using less computational cost**. In addition, table 2 demonstrates that SmartRAG refuses to retrieve when the database contains too few information such that it can save computational cost. We will emphasize this in a revised version.
>
> Of course this inference advantage comes at the cost of an additional RL training phase. We think that is a reasonable sacrification to save inference cost. We will also discuss the tradeoff in the our revision.
>
> **4.  Considering the potential for optimizing PPO's efficiency or exploring alternative RL algorithms to reduce training costs.**
>
> The reviewer made a great point that better RL algorithms may lead to even better performance and efficiency for our method.
>
> Actually we also noticed this aspect. In this paper, we primary discussed about different initial policies for RL in section 4.4. We did not discuss the reinforcement learning potential in a more systematic way due to the page limit and we leave it for future work. Of course we will add a section to briefly discussing the potentials, as the reviewer has pointed out.

---

> ### Author Response · Authors · 2024-11-21
> **Paper revision**
>
> Following the reviewer's suggestion, we have revised the paper. Here are the details:
>
> - We add a new subsection named *Advantage of Joint Optimization* (section 4.1) to explicitly discuss the advantage of SmartRAG comparing to separately optimizing individual modules. The original Section 4.1 is now renumbered as Section 4.2, and subsequent sections are adjusted accordingly.
>
> - We add three more baselines in table 1 and two more baselines in figure 2.
>
> - A new subsection (section A.5) discussing the training and inference latency is included.
>
> Further suggestions to improve the quality of the paper are also welcomed!

---

### Meta-Review · Area_Chair_qD8z · 2024-12-08

**Metareview:**

This paper introduces SmartRAG, a retrieval-augmented generation (RAG) framework that improves performance by jointly optimizing its components. Unlike conventional RAG methods, SmartRAG employs a policy network to dynamically determine when to retrieve information, what queries to issue, and how to synthesize responses from retrieved observations. The framework leverages reinforcement learning with environment feedback to train the system holistically, enhancing module coordination and reducing retrieval costs. Experimental results on multiple QA datasets demonstrate that SmartRAG outperforms systems with independently optimized components. The authors have added more baselines in the response, and addressed other key concerns raised by the reviewers.

**Additional Comments On Reviewer Discussion:**

NA

---

### Decision · Program_Chairs · 2025-01-22

Accept (Poster)